# Improving Plasmonic Photothermal Therapy of Lung Cancer Cells with Anti-EGFR Targeted Gold Nanorods

**DOI:** 10.3390/nano10071307

**Published:** 2020-07-03

**Authors:** Oscar Knights, Steven Freear, James R. McLaughlan

**Affiliations:** 1School of Electronic and Electrical Engineering, University of Leeds, Leeds LS2 9JT, UK; O.B.Knights@leeds.ac.uk (O.K.); s.freear@leeds.ac.uk (S.F.); 2Leeds Institute of Medical Research, St James’ University Hospital, University of Leeds, Leeds LS9 7TF, UK

**Keywords:** nanoparticles, gold nanorods, cancer therapy, photothermal therapy, photoacoustic imaging, lung cancer, EGFR-targeting

## Abstract

Lung cancer is a particularly difficult form of cancer to diagnose and treat, due largely to the inaccessibility of tumours and the limited available treatment options. The development of plasmonic gold nanoparticles has led to their potential use in a large range of disciplines, and they have shown promise for applications in this area. The ability to functionalise these nanoparticles to target to specific cancer types, when combined with minimally invasive therapies such as photothermal therapy, could improve long-term outcomes for lung cancer patients. Conventionally, continuous wave lasers are used to generate bulk heating enhanced by gold nanorods that have accumulated in the target region. However, there are potential negative side-effects of heat-induced cell death, such as the risk of damage to healthy tissue due to heat conducting to the surrounding environment, and the development of heat and drug resistance. In this study, the use of pulsed lasers for photothermal therapy was investigated and compared with continuous wave lasers for gold nanorods with a surface plasmon resonance at 850 nm, which were functionalised with anti-EGFR antibodies. Photothermal therapy was performed with both laser systems, on lung cancer cells (A549) in vitro populations incubated with untargeted and targeted nanorods. It was shown that the combination of pulse wave laser illumination of targeted nanoparticles produced a reduction of 93%±13% in the cell viability compared with control exposures, which demonstrates a possible application for minimally invasive therapies for lung cancer.

## 1. Introduction

Cancer is a leading cause of death worldwide with approximately 70% of deaths occurring in low- and middle-income countries [1]. Lung cancer is the most prevalent and deadly form of cancer since there exists very few options for diagnosis or treatment. Plasmonic photothermal therapy (PPTT) is a therapeutic modality, when combined with AuNRs could provide a highly selective, minimally invasive treatment option for cancer. It would be beneficial if PPTT could be administered with a pulsed-wave (PW) laser since it would reduce the potential damage to surrounding tissues by eliminating the bulk heating effect caused by continuous wave (CW) lasers. The photothermal effect relies heavily on a light source that can deliver sufficient of energy to a localised region, and thus a laser is often used [2]. Laser ablation (LA), a common clinical therapeutic technique that relies on lasers, is predominantly used to compliment additional therapies by reducing tumour volume [3]. It is mostly used for treating superficial and lung cancers where laser access and light delivery is feasible [4]. Continuous wave (CW) lasers with high powers (around 5 W) are employed to induce bulk heating and irreversible thermal damage in the target tissue; however, pulsed wave (PW) lasers have also shown potential for photothermal applications [5]. Depending on the type of laser system employed—either PW or CW—there will be significant differences in the observed outcomes. CW lasers can induce either apoptosis or necrosis, depending on laser intensity and AuNR distribution, whereas PW lasers can only induce necrosis [6]. These two pathways for cell death have their own advantages and disadvantages. For example, an apoptotic pathway can lead to cells developing drug and thermal resistance but does not cause immunogenic or inflammatory responses, while the opposite is true for a necrotic pathway [7]. PW lasers create a highly-localised rapid temperature increase in the target AuNRs [8], and this almost-instantaneous and high temperature increase causes large mechanical stresses (peak pressures of 10–100 MPa [9]) that can induce necrotic cell death depending on particle location and laser energy. There are very few reports on pulsed wave plasmonic photothermal therapy (PW-PPTT), also known as photoacoustic plasmonic photothermal therapy (PA-PPTT), and the majority predominately use either high energy laser pulses, ultra-short laser pulses (femtosecond), or alternative photoabsorbers, with little in the way of low-energy, nanosecond pulses that utilise AuNRs as the absorbing agent. Moreover, there are few reports addressing how the size of the AuNRs may affect the treatment efficacy of both PW-PPTT and conventional PPTT at equivalent concentrations. The optimisation of both the optical absorbers and laser parameters is crucial to the success of this technique. If PW lasers can be used to destroy target regions of tissue successfully and efficiently, with similar or superior outcomes to that of CW lasers, then new and combined diagnostic and therapeutic techniques may be possible.

The functionalisation of AuNRs to molecularly target specific binding sites, such as epidermal growth factor receptors (EGFR), is increasingly seen as an essential aspect of using AuNRs for biomedical purposes. This is largely due to the need for high numbers of AuNRs to be localised in a tumour region for a sufficient PA or PPTT effect to be observed. Furthermore, relying solely on the enhanced permeability and retention (EPR) effect to accumulate AuNRs in target tissue may not be sufficient [10,11]. If the target ligand is known, then the AuNRs can be functionalised with monoclonal antibodies (for example anti-EGFR) that will enable monovalent affinity. This is a highly desirable characteristic that can result in a much larger accumulation of AuNRs at a site. It is known that many forms of cancer express EGFR-positive ligands and it has therefore become a common method for molecular targeting for a range of imaging techniques such as photoacoustic imaging PAI [12]. The aim of this study was to investigate the effects that AuNR targeting to EGFR positive lung cancer cells has on both CW and PW laser treatment.

## 2. Materials and Methods

To determine the EGFR expression of lung cancer cells, immunofluorescence (IF) staining was performed using a standard IF protocol. Briefly, the A549 cells were grown in a 6-well plate on microscope coverslips. Once 70% confluence was reached, the media was removed and the cell monolayer was washed with Dulbecco’s Phosphate-Buffered Saline (DPBS, Thermo Fisher Scientific, Waltham, MA USA). To fix the cells to the coverslips, 4% PFA (Paraformaldehyde, Thermo Fisher Scientific, Waltham, MA USA) was added and left for 15 min at room temperature. The coverslips were then washed twice with DPBS before being permeabilised with a solution of DPBS and 0.3% Triton X-100. The coverslips were then washed twice in DPBS, followed by sample blocking in 10% FBS (fetal bovine serum, Thermo Fisher Scientific, Waltham, MA USA) for 1 h. The blocking buffer was removed and Alexa Flour 488-conjugated anti-EGFR antibodies, diluted in 5% FBS, was added to the coverslips and incubated for 2 h at room temperature. Finally, the coverslips were washed 3 times in DPBS and mounted on microscope slides using DAPI (4′,6-diamidino-2-phenylindole, Thermo Fisher Scientific, Waltham, MA USA) reagent (ProLong Gold Antifade Mountant). The same protocol was repeated to form a control group without adding the conjugated antibodies. The level of EGFR expression (as determined from the IF images in Figure 1) in A549 cells was not as high as expected. Nevertheless, EGFR expression was observed during the IF staining and therefore it was decided that the effect of targeting AuNRs to the EGFR receptors would be investigated.

AuNR-targeting was considered using dark-field microscopy. Highly concentrated Streptavidin-conjugated AuNRs were purchased (Table 1) from Nanopartz—a company that is able to synthesis AuNRs on a large scale with repeatable characteristics—to enable them to be easily functionalised with an anti-EGFR targeting ligand (Figure 2). A protocol was designed to enable the comparison of cellular uptake for targeted versus untargeted S-Au40-849s at three different time-points: 4 h, 8 h, and 24 h. These were chosen to provide an indication of the timescales required for wide-spread affinity of the targeted AuNRs to the lung cancer cells. We previously observed high levels of uptake after 4 h incubation [13]. The biotinylated antibody used in this study was a mouse monoclonal antibody (ab24293, abcam) and the amount required to facilitate the conjugation with the AuNRs was explored along-side the targeting efficiency. A series of 21 microscope coverslips were soaked in ethanol overnight before being allowed to dry completely and were then placed inside a separate well of six 6-well plates. For each time-point there was enough cover-slips to allow for: one blank (3), two untargeted (6), two targeted with 10 μL biotinylated antibody (6), and two targeted with 20 μL biotinylated antibody (6). A human non-small cell lung epithelial carcinoma cell line (A549, ATCC, Middlesex, UK) was cultured in DMEM (Dulbecco’s Modified Eagle Medium) media supplemented with 10% FBS (Fetal Bovine Serum). When the cells reached 80% confluency, the 6-well plates were seeded with $4\times 10^{3}$ cells per well and incubated for 48 h. Before the AuNRs were added to the wells, a functionalising protocol was followed (Figure 3). The stock S-Au40-849s (concentration = 13.2
mg
m$L^{-1}$) were first sonicated for 15 min to minimise aggregation and then 45 μL of the stock S-Au40-849s was split between three sterile Eppendorfs (i.e., 15 μL per Eppendorf). The first Eppendorf (named ‘UT’) was reserved for untargeted AuNRs, 10 μL of the biotinylated antibody was added to the second Eppendorf (named ‘T10’), and 20 μL of the biotinylated antibody was added to the final Eppendorf (named ‘T20’). All of the Eppendorfs were then topped up with Dulbecco’s phosphate-buffered saline (DPBS, 14190-094, Thermo Fisher Scientific, Waltham, MA USA) to a total volume of 40 μL (i.e., 25 μL DPBS in Eppendorf UT, 15 μL DPBS in Eppendorf T10, and 5 μL DPBS in Eppendorf T20). All three Eppendorfs were then sonicated for 10 min before being placed on a vortex for 30 min to facilitate the conjugation of the AuNRs and antibodies. The AuNRs were then purified by centrifuging for 10 min at a relative centrifugal force (r.c.f.) of 5900, followed by the removal of the supernatant and addition of 40 μL DPBS, and then further mixing was performed with the vortex for 2 min followed by sonication for 10 min. After repeating the purification process twice, the supernatant was replaced with 100 μL DMEM media before being added to separate 15 mL falcon tubes filled with DMEM media to a total volume of 6 mL. Finally, each falcon tube was vortexed for 2 min and sonicated for 10 min before the media-AuNR solutions were added to the 6-well plates at a total volume of 2 mL per well, giving a final AuNR concentration of 30 μg/mL. The plates were then incubated for different lengths of time ( 4 h, 8 h, and 24 h) before a standard protocol for preparing microscope slides for dark-field imaging was followed. A549 cells were plated onto 22×22 mm glass cover-slips in a 6-well plate at a density of $1\times 10^{5}$
${\mathrm{well}}^{-1}$ and allowed to grow for two days. The AuNR-media was removed from each well and the cell monolayer on the cover-slip was twice-rinsed with DPBS, fixed in 4% paraformaldehyde/DPBS for 10 min at room temperature and rinsed with DPBS five times. The fixed coverslips were then mounted and sealed onto glass slides. Bright and dark-field microscopy imaging was performed with an inverted microscope (Nikon Eclipse Ti-E, Nikon UK Ltd., Kingston upon Thames, Surrey, UK) using an oil coupled 100x objective (CFI Plan Fluor, Nikon UK Ltd., Kingston upon Thames, Surrey, UK). Images were recorded with a 5 Megapixel colour camera (DS-Fi1, Nikon UK Ltd., Kingston upon Thames, Surrey, UK) and saved using the NIS-Elements D software (Nikon UK Ltd., Kingston upon Thames, Surrey, UK). The height of objectives focal plane was monitored to establish that images were acquired within the cells. Open-source software package ImageJ [14] was used to crop and enhance the contrast of saved images. To ensure valid comparisons could be made, all of the images were enhanced in the same way.

For the photothermal therapy experiments, targeting was achieved using the same methodology (Figure 3) for conjugating the S-Au40-849s to anti-EGFR monoclonal antibodies was followed as described. However, the 6-well plates were replaced with two 96-well plates and were seeded with A549 cells at a concentration of $1\times 10^{5}$ cells per well. The experimental procedure was similar to that previously described [13], with untargeted (UT) AuNRs were add to one plate and targeted (T10) AuNRs were added to the other—both at a concentration of 30 μg
m$L^{-1}$—and left to incubate for either 4 h or 24 h. Immediately prior to laser exposure, the media in the wells of the 96-well plate was removed, the cell monolayer washed once with DPBS, and then 100 μL of fresh media was added. This washing step was performed to minimise the number of AuNRs remaining in the wells that were not bound to the cell surface receptors or taken up by the cells. The laser systems used were a pulsed tuneable laser (Surelite, OPO Plus, Continuum, San Jose, CA, USA) operating at a pulse repetition frequency of 10 Hz with a pulse duration of 7 ns, spot size of 9 mm (at the bottom of the 96-well plate) and radiant exposure of 25 mJ
c$m^{-2}$, and a continuous wave diode laser (B4-852-1500-15C, Sheaumann Laser, Marlborough, MA, USA) operating at 1.5
W across a spot size of 9 mm with a fixed wavelength at 854 nm. The pulsed laser system was tuned to the same wavelength as the CW laser ( 854 nm) to ensure an accurate comparison could be made between the two laser types, by facilitating an equivalent optical absorption by the AuNRs. This wavelength was confirmed using a direct measurement of the beam with a UV-VIS-NIR spectrometer (HR4000, Ocean Optics, Orlando, FL, USA). Output from either laser was coupled directly into broadband optical fibres that were mounted onto a 3-axis motorised translation stage [13] to enable the scanning of the fibre tips across the 96-well plates. The wells that were targeted with the lasers were alternated to reduce any effects from the laser heating of neighbouring wells, and each well was exposed for a total of 5 min.

To measure cell viability, a standard MTT (3-[4,5-dimethylthiazol-2-yl]-2,5 diphenyl tetrazolium bromide) colorimetric assay protocol was followed [15] to establish the level of cell death between the two laser types. This metabolic assay provides an indication of cell viability by measuring the enzymatic activity of cellular mitochondria [16]. After laser irradiation, the media from each well of the 96-well plate was replaced with fresh media and the plate was placed in the incubator. 24 h later, the media was removed from each well and a solution of media containing MTT ( 500 μg
m$L^{-1}$) was added. After a further 3.5
h incubation the media containing MTT was removed from each well and the 96-well plate was wrapped in foil and stored at approximately 4 ${}^{{}^{\circ}}C$ ready for absorbance measurements. Before measuring the plates with a plate reader (Mithras LB 940, Berthold Technologies, Bad Wildbad, Germany), 100 μL DMSO was added to each well. The divided sections of the plate were averaged to obtain a single absorbance value for each AuNR concentration, and the background absorbance level was subtracted from each of the other values. The cell viability was finally calculated by the ratio of mean absorbance of the sample with respect to mean absorbance of the control group (cells with media and no laser or AuNR exposure).

## 3. Results

Bright and dark-field imaging was used to assess the targeting of AuNR clusters to the lung cancer cells. Several images (N =248) were acquired by scanning pseudo-randomly across the microscope slides to cover the majority of the sample and ensure a representative illustration of the cellular uptake was observed. Figure 4 shows a selection of those images to demonstrate the typical distribution observed. The volume of biotinylated antibodies ( 10 μL or 20 μL) that was used in the conjugation process appeared to affect the aggregation of AuNRs. As can be seen from Figure 5, large aggregates formed when 20 μL biotinylated antibodies was used in the conjugation process. The AuNR aggregates did appear to be bound to the surface of the A549 cells, however the large number of AuNRs making up the aggregates lead to a decrease in the distribution of AuNRs across the cell sample. When 10 μL was used, a reduction in AuNR aggregates was seen and the overall distribution of AuNRs throughout the cell population was more uniform.

Figure 6 and Figure 7 show the cell viability data of the A549 cells after incubation with both untargeted and targeted S-Au40-849s for 4 h and 24 h, respectively. After a 4 h incubation period, there was no discernible reduction in cell viability induced from the AuNRs, either on their own or following laser irradiation. The lack of photothermal ablation can be attributed to an insufficient number of AuNRs remaining in the absorbing region after washing. This is in agreement with the cellular uptake data (Figure 4) where a 4 h incubation period resulted in minimal uptake of both untargeted and targeted S-Au40-849s. Conversely, after 24 h incubation with S-Au40-849s (Figure 7), reduced cell viability was observed in some cases. The AuNRs that received no laser exposure did not reduce the viability of the lung cancer cells, independent of whether they had been functionalised with the anti-EGFR ligands.

## 4. Discussion

The untargeted S-Au40-849s displayed little cellular uptake across all of the time points, which was surprising since it might be expected that after 24 h incubation, much higher levels of uptake would be seen compared to that which was observed here. In comparison with the dark-field images taken of the citrate-capped AuNRs used in [13,17], there was considerably less cellular uptake of the untargeted S-Au40-849s. This was likely due to the streptavidin ligands that were already conjugated to the surface of the AuNRs, limiting the penetration of the S-Au40-849s into the lung cancer cells, since streptavidin has a relatively large molecular weight (approximately 60 kDa) and has been shown to restrict cellular uptake [18]. Furthermore, the potentially reduced biocompatibility of untargeted S-Au40-849s may negatively affect cellular uptake. Conversely, the targeted S-Au40-849s significantly enhanced AuNR uptake by the lung cancer cells after 8 h and 24 h (potentially via affinity to EGFR receptors) and while minimal uptake was observed after 4 h incubation with the A549 cells, there was an increase in overall uptake compared with their untargeted counterparts. The conjugation of the S-Au40-849 AuNRs with the anti-EGFR monoclonal antibodies had a considerable effect on the overall uptake when incubated with the lung cancer cells for longer than 4 h and the results provide a compelling argument to use molecularly targeted AuNRs for the selective delivery of high concentrations of AuNRs to malignant tissues. This was in contrast to a previous study [13] where a small but notable reduction in cell viability (approximately 20%) was observed. Differences between these studies can be attributed to the differences in the surface chemistry of the AuNRs used. In this study, the AuNRs were either surrounded by streptavidin proteins—which reduce cellular uptake (Figure 4) and therefore reduce toxic effects—or were conjugated with anti-EGFR ligands and were bound to the surface receptors on the cell membrane, limiting penetration into the cells. The cells that experienced a combination of untargeted S-Au40-849s and either CW or PW laser irradiation, also did not display a reduction in cell viability after the 5 min laser exposure. As discussed previously, the untargeted AuNRs had limited penetration into the cell membranes due to the large streptavidin molecules bound to the surface of the AuNRs and so were likely almost entirely removed during the washing stage.

Perhaps the most significant result was that the PW laser reduced the viability of 93%±12% of the population of lung cancer cells when combined with anti-EGFR targeting S-Au40-849s (after 24 h incubation), in comparison to the CW which destroyed almost half (46%) of the cells. A direct comparison of the energy/power density of these two laser systems is difficult due to their different modes of operation (see Appendix A for further discussion), but by just taking average power, gives an $P_{\mathrm{ave}}=160\;\mathrm{mW}$ for the PW laser and 950mW or the CW laser. There are likely multiple contributing factors to the enhanced PPTT efficacy of the PW laser. The first reason may be that the efficacy of the CW laser for inducing cell-death is at its highest when there are a large number of AuNRs in the absorbing region and the light can be efficiently converted into wide-spread, bulk heating. The washing of the cell monolayer removed any AuNRs that were not bound to the cell surface and the number remaining in the path of the laser will not have been high enough to induce hyperthermia. The AuNRs that did persist would still have absorbed the incoming laser-light and converted it into heat, causing damage to the cells, however the overall heat generated from the exposure was not enough to provide a broad destruction of cells. The second reason may be due to an enhanced optical absorption and bubble-formation around the AuNRs under PW exposure due to AuNR clustering. It has been shown that the accumulation of antibody conjugated AuNRs on the surface of a cell membrane can further facilitate the self-assembly of the AuNRs into nanoclusters [19]. This in turn leads to an enhancement of the bubble formation around the AuNRs under high intensity laser pulses and subsequently an increase in damage to the cellular membrane. Zharov et al. (2005) concluded that pulsed lasers were more effective at inducing cell death when AuNRs formed nanoclusters on the cell membrane, and this agrees with our findings. Highly localised bubble formation due to the presence of nanoparticles can have other benefits for both therapeutic and imaging applications [20,21].

## 5. Conclusions

CW lasers induce bulk temperature changes in an absorbing medium whereas localised heating occurs when PW lasers are used. This suggests that PW lasers are only able to destroy cancer cells via a necrotic pathway, which reduces the risk of the cells developing drug or heat resistance but can cause immunogenic or inflammatory responses due to the immediate expulsion of the cells internals. Conversely, depending on the laser exposure parameters, CW lasers have been shown in the literature to induce cell death via either apoptosis or necrosis (although this aspect was not addressed in this study). Understanding these important distinctions between the two laser types is imperative for guiding therapy design. The relevant temperatures required to initiate the destruction of cells and the inducement of hyperthermia is in excess of 42 ${}^{{}^{\circ}}C$, which is only approximately 5 ${}^{{}^{\circ}}C$ above internal body temperature. The length of time the cells remain at elevated temperatures influences the therapeutic efficiency and a clinically relevant parameter known as the thermal isoeffective dose (TID) is often used to compare thermal treatments. There is a large range of TID threshold values in the literature for determining when complete destruction of tissue is achieved, and this is due to the differences between the cellular compositions of different tissues. If TID values are to be used as a clinical measure of therapy, it is necessary to determine a TID threshold for the specific tissue that is being treated, otherwise it may be difficult to ensure maximum therapeutic effect.

Further to the consideration of AuNR size on photothermal efficacy, the effect of targeting AuNRs to the lung cancer cells was demonstrated. AuNRs with a similar size and aspect ratio to those used previously were purchased with streptavidin proteins conjugated to the surface, enabling the AuNRs to be easily functionalised with anti-EGFR targeting ligands. Lung cancer cells (along with many other malignant tissues, [22]) have been shown to overexpress anti-EGFR receptors and provide a potential method for increasing AuNR delivery efficiency to the cells—a critical aspect to the success of this therapy. The cellular uptake of untargeted S-Au40-849s was compared with that of targeted ones, and the results demonstrated the importance of molecularly targeting AuNRs for increased uptake. The untargeted AuNRs showed minimal uptake across the cell samples after the longest incubation time studied ( 24 h), however the accumulation and uptake of targeted AuNRs was evident. Under laser irradiation, targeting also provided a significant advantage over untargeted AuNRs, when the incubation time was long enough ( 24 h). After 4 h there was no observed reduction in cell viability for any of the parameters studied, which agreed with the uptake study. However, a 24 h incubation resulted in an observable reduction in cell viability in some cases. The untargeted AuNRs were not able to reduce cell viability under any laser exposure parameters. In the case for targeted AuNRs, the PW laser was the most efficient laser system, destroying almost 93% of the total population of lung cancer cells, compared with almost half for the CW laser. This reversal in photothermal efficacy between the laser types may be due to the AuNRs forming nanoclusters, facilitated by the anti-EGFR targeting ligands, and increasing the mechanical stresses induced by the PW laser. In terms of the therapeutic efficacy of different sized un-targeted AuNRs, it appears that smaller AuNRs (width =10 nm) are most suited as photoabsorbers under both CW and PW laser illumination, whereas AuNRs with a width of 25 nm were the least suited. The location of AuNRs was a critical aspect for therapeutic efficacy and targeting them to bind specifically to cell-receptors, such as EGFR, can improve cellular uptake and overall therapeutic outcome. Furthermore, if the eventual goal is to use AuNRs as clinical therapeutic agents then it is crucial that the production of AuNRs with consistent and uniform dimensions, properties and coatings, can be scaled-up to a level that ensures the practicality and safety of wide-spread clinical use.

## Figures and Tables

**Figure 1 nanomaterials-10-01307-f001:**
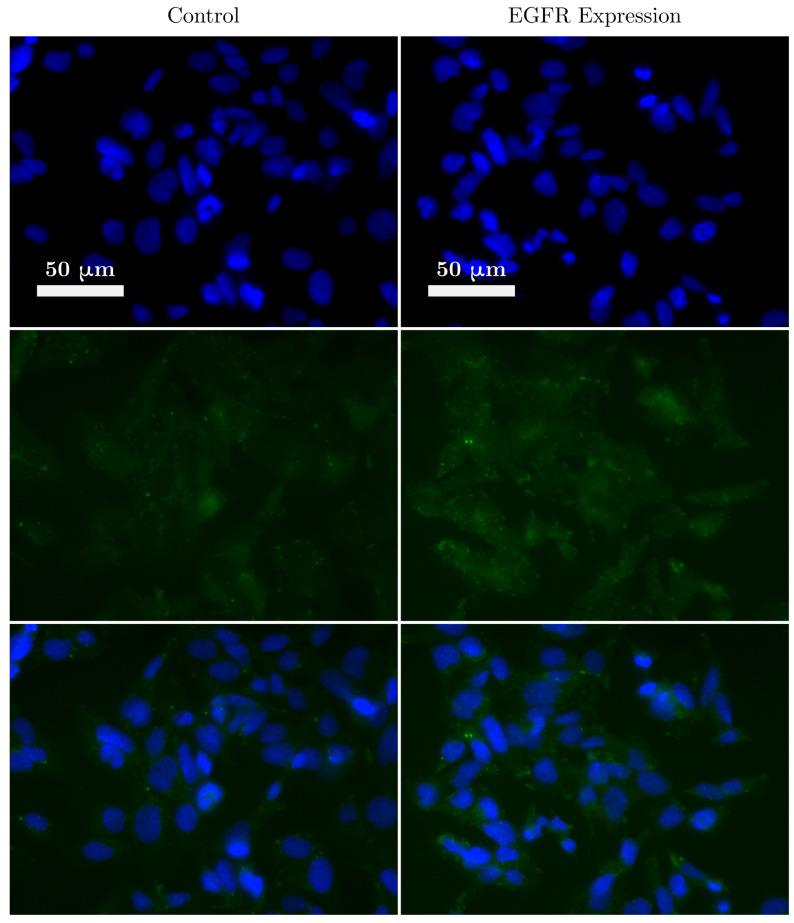
A fluorescence image of lung cancer cells (A549) showing the expression of anti-epidermal growth factor receptors receptors. Blue represents the staining of the cell nucleus and green shows the expression of EGFR.

**Figure 2 nanomaterials-10-01307-f002:**
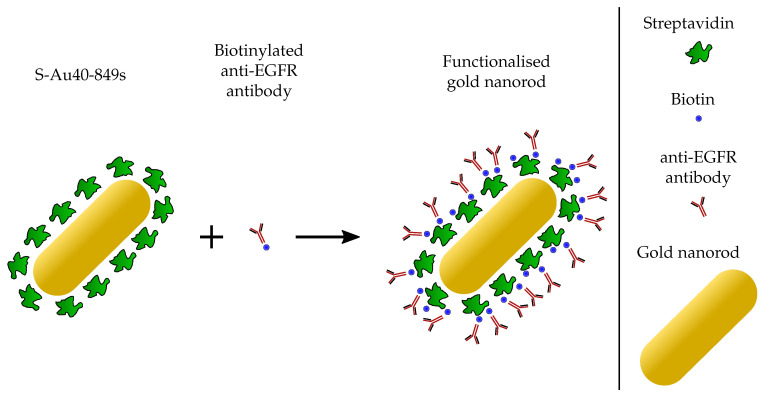
The biotinylated anti-EGFR antibodies exhibit high affinity to the streptavidin proteins that are already conjugated to the AuNR.

**Figure 3 nanomaterials-10-01307-f003:**
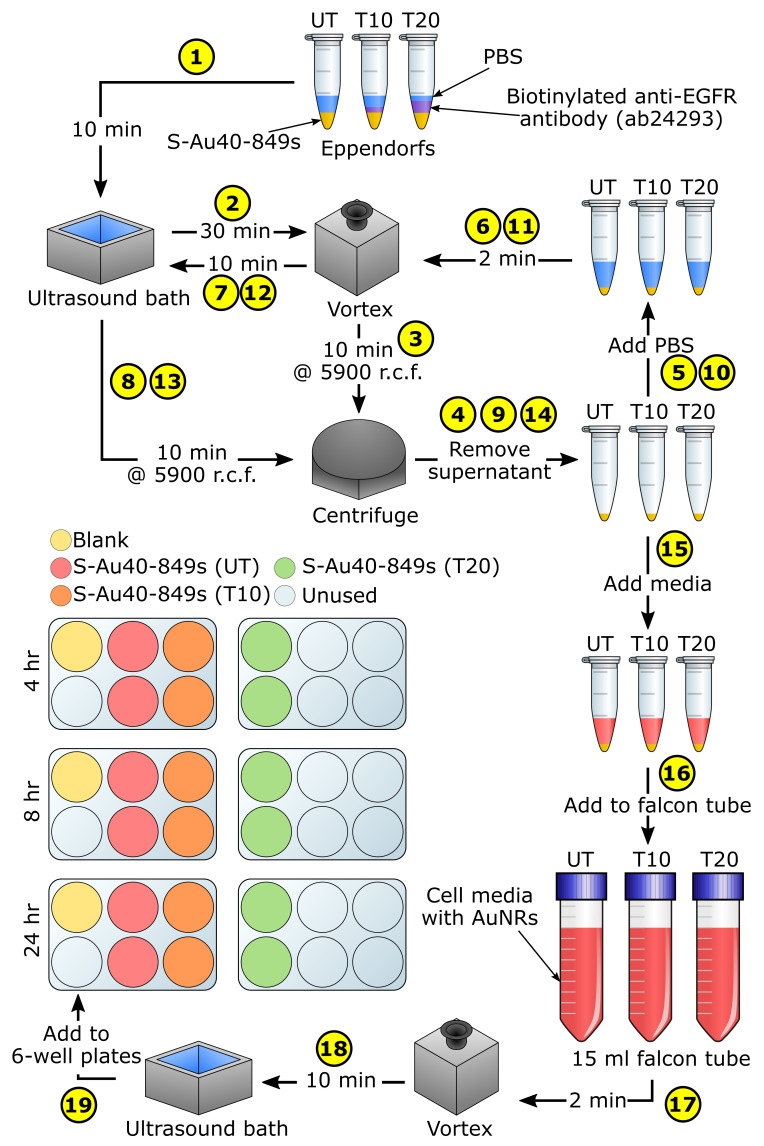
A schematic depicting the functionalisation process for the S-Au40-849s. UT = untargeted AuNRs, T10 = targeted AuNRs ( 10 μL antibody), T20 = targeted AuNRs ( 20 μL antibody).

**Figure 4 nanomaterials-10-01307-f004:**
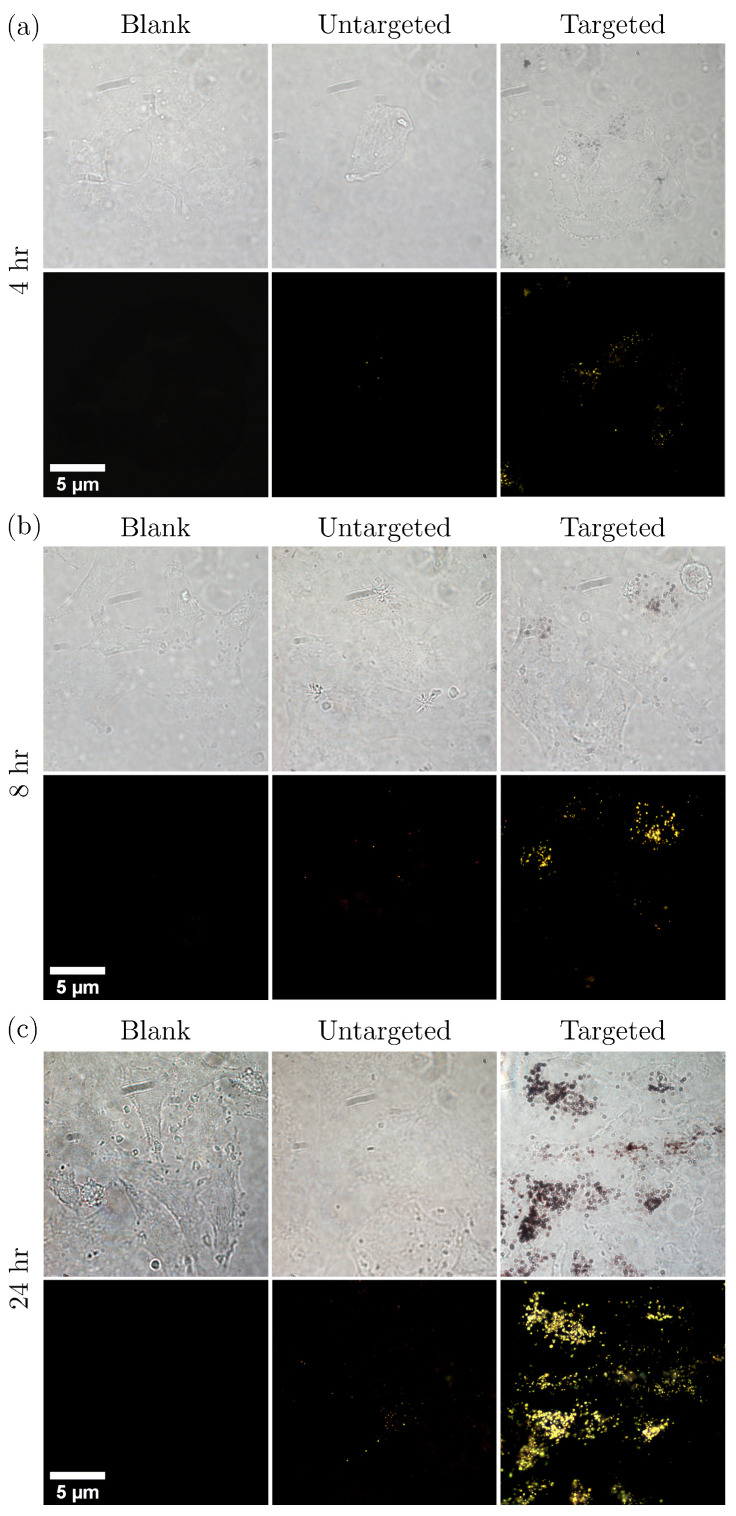
Bright- and dark-field images of lung cancer cells (A549) incubated with untargeted versus targeted ( 10 μL antibody) S-Au40-849s for (**a**) 4 h, (**b**) 8 h, and (**c**) 24 h.

**Figure 5 nanomaterials-10-01307-f005:**
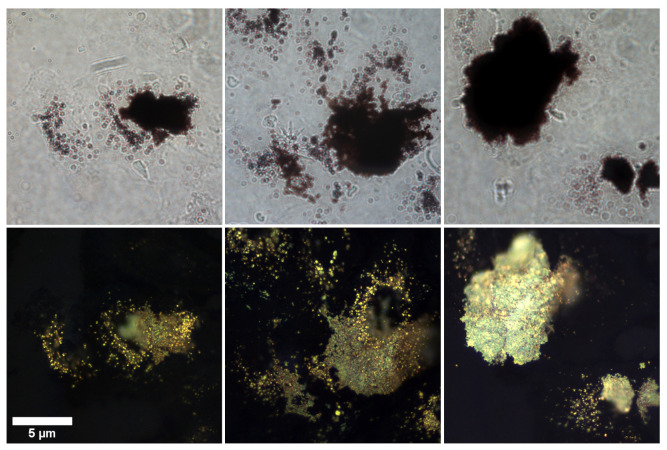
Large AuNR aggregates were apparent when a high volume ( 20 μL) of biotinylated anti-EGFR antibodies were used in the conjugation process of the S-Au40-849s, after 24 h of incubation.

**Figure 6 nanomaterials-10-01307-f006:**
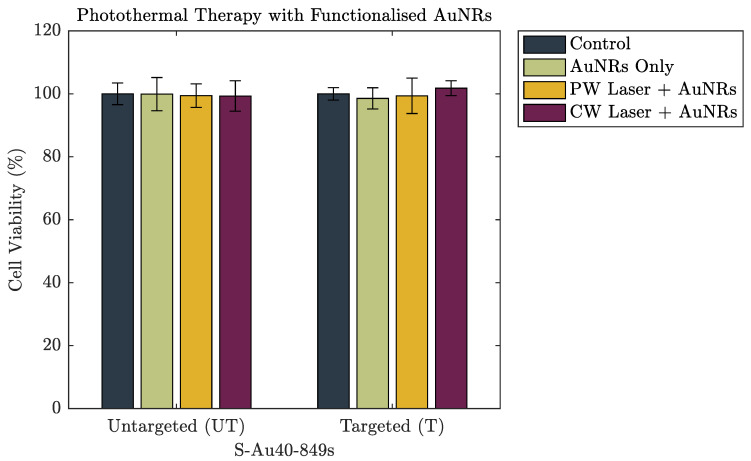
The cell viability of a lung cancer cell line after 4 h incubation with either untargeted S-Au40-849s or targeted S-Au40-849s.

**Figure 7 nanomaterials-10-01307-f007:**
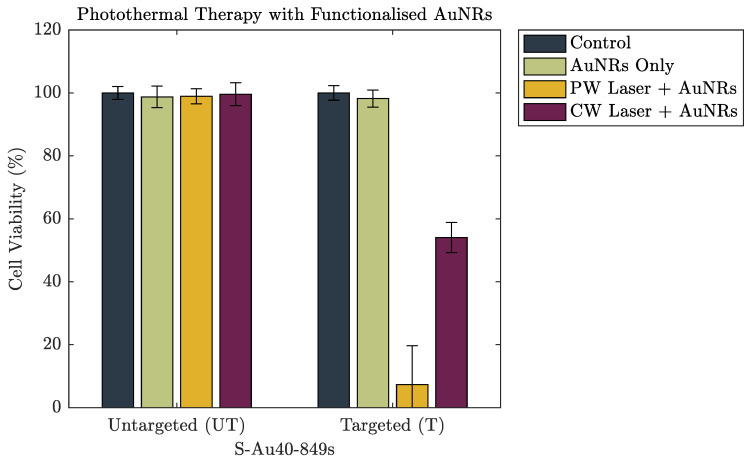
The cell viability of a lung cancer cell line after 24 h incubation with either untargeted S-Au40-849s or targeted S-Au40-849s.

**Table 1 nanomaterials-10-01307-t001:** Certified dimensions and measured surface plasmon resonance of all AuNRs used in study.

AuNR Name	Width (nm)	Length (nm)	Aspect Ratio	SPR (nm)
S-Au40-849	40	148	3.7	849

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
