# Peer review of "Improving Plasmonic Photothermal Therapy of Lung Cancer Cells with Anti-EGFR Targeted Gold Nanorods"

_nanomaterials, 2020, doi:10.3390/nano10071307_

Round 1

Reviewer 1 Report

The authors describe two approaches for improving photothermal therapy in a human lung cancer cell line. Although the targeting approach, using anti-EGFR antibodies, has been described by a number of investigators, the efficacy of pulsed laser irradiation in photothermal therapy is less well known and, as such, the manuscript is likely to be of interest to researchers in the photothermal therapy/nanotherapeutics fields.  

As stated on page 2, EGFR expression in this cell line was rather poor and, as such, the authors should have considered using another cell line with better EGFR expression. It would have been useful to compare several cancer cell lines known to express EGFR. Please discuss briefly.

The cell viability assay (Figs. 6 and 7) is not described? Please include a detailed description in the Materials and Methods section.

In the discussion section, the authors discuss the increased uptake of the targeted nanorods by A549 cells, however, uptake suggests that the targeted AuNR are internalized within the cells. Do the authors have any evidence that the targeted AuNRs are internalized as opposed to being bound to the EGFR on the membrane? If internalized, this would suggest receptor-mediated endocytosis? If not internalized, perhaps the authors should not use “uptake,” but rather “bound” or “binding.” Please comment.

  1. 1, line 7: “gold nanorods that have accumulated…”
  2. 1, line 24: “when combined with gold nanorods (AuNRs), can provide…”
  3. 4, line 88: “high levels of uptake after 4h…”
  4. 5, line 136-137: The authors state: “Both of the laser systems were operating under the same parameters as in the previous section…” It’s not clear what section is referred to. Please clarify. Also, I assume that the irradiation wavelength of the two laser systems (PW and CW) were the same? Please state the wavelength and the type(s) of laser system(s) used.

Figure 4: I assume that the images were acquired using the low concentration (10 mL) formulation? Please state explicitly in the figure caption.

Figure 5: I assume a 24 h incubation period was used? Please state explicitly in the figure caption.

Reviewer 2 Report

The manuscript written by Knights et al. focuses on the application of targeted AuNRs as photothermally active agents to reduce the viability of lung cancer cells. The Authors claim, that in contrast to the CW lasers, pulsed wave laser irradiation proved to be more effective in combination with EGFR-targeted AuNRs. The paper is well-written, and the experiments had been planned appropriately. Nevertheless, I found missing information and parts which are not described sufficiently in the present form of the manuscript. Consequently, I would suggest the acceptance of the paper only after addressing my major concerns:

  1. What considerations led the Authors to choose the dimension of the AuNRs? This basically the most important factor that determines the outcome of the experiments, because the penetration-abilities of the NRs highly depend on the size (AR). Have the Authors done any modification on the purchased rods prior to the surface-modification? (as I checked the website of Nanopartz, the initial concentration of these NRs is 2500 ug/mL, which is more than 5 times lower than the written concentration in Line 98).
  2. How was it possible to precipitate the NRs at 5.9 g (=RCF)? Has this value been miswritten? Please comment on this, because if the particles could be precipitated at this force, they were certainly aggregated.
  3. Please specify the optical setup more in detail. Laser wavelength is missing, although this is one of the most important parameters to be considered regarding NR size, LSPR, penetration depth, etc. Which plasmonic band has been excited: the transversal or the longitudinal one? What was the repetition rate and the pulse duration for PW irradiation? Was the laser coupled to the microscope, or is it an external illumination setup? How does the CW power correspond to the power density of PW (are they comparable in power)?
  4. What does the Author think: would not be more efficient to use NRs with a longitudinal resonance in the NIR wavelength region? This would ensure the penetration of the laser through the skin in later in-vivo human applications.
  5. Are the Authors sure, that the particles do not aggregate after the addition of biotinylated antibodies even in case of 10uL? Some optical data (extinction spectra, zeta potential, etc.) would be necessary to prove the existence of individual particles, because the cellular uptake of large aggregated differs significantly. On the other hand, aggregates behave differently upon illumination: aggregation, clustering might broaden the LSPR dramatically.
  6. What is the difference between cellular uptake and surface adsorption, and could these be distinguished based on microscope images?
  7. Does Figure 4 belong to 10uL biotinylated antibodies? Please specify it in the figure caption.
  8. How did the Authors determine the cell viability?
  9. The Authors explain the smaller extent of penetration of the non-targeted AuNRs with the streptavidin ligand shell. However, the targeted ones have also the same streptavidin shell, which is used for anchoring the EGFR-antibodies. If the AuNRs with solely streptavidin cannot penetrate into the cell (the term ‘uptake’ is of great importance here as well), why can the biotinylated ones penetrate?
  10. Please explain the major differences between CW and PW irradiation more in detail (Line 197). How does this ‘bubble formation’ mechanism look like?
  11. The Authors write in Line 240-241, that smaller NRs might have even better efficiency. Could you please comment on the decision, why such long NRs have been chosen in your study? Concentration of NRs also alter the optical density, hence the photoabsorption of the sample. Could the increase of the AuNR number/cell further enhance their photothermal efficiency (is there any upper limit)?

Round 2

Reviewer 2 Report

Authors addressed all my concerns in the revised version of the manuscript. In the present form, I accept it and recommend to publish it in the journal. Thank you.